# Position: Beyond Assistance – Reimagining LLMs as Ethical and Adaptive Co-Creators in Mental Health Care

**Abeer Badawi** [1 2]  **Md Tahmid Rahman Laskar** [3 4]  **Jimmy Xiangji Huang** [3]  **Shaina Raza** [2]  **Elham Dolatabadi** [1 2]

## Abstract

This position paper argues for a fundamental shift in how Large Language Models (LLMs) are integrated into the mental health care domain. We advocate for their role as co-creators rather than mere assistive tools. While LLMs have the potential to enhance accessibility, personalization, and crisis intervention, their adoption remains limited due to concerns about bias, evaluation, over-reliance, dehumanization, and regulatory uncertainties. To address these challenges, we propose two structured pathways: SAFE-*i* (Supportive, Adaptive, Fair, and Ethical Implementation) Guidelines for ethical and responsible deployment, and HAAS-*e* (Human-AI Alignment and Safety Evaluation) Framework for multidimensional, human-centered assessment. SAFE-*i* provides a blueprint for data governance, adaptive model engineering, and real-world integration, ensuring LLMs align with clinical and ethical standards. HAAS-*e* introduces evaluation metrics that go beyond technical accuracy to measure trustworthiness, empathy, cultural sensitivity, and actionability. We call for the adoption of these structured approaches to establish a responsible and scalable model for LLM-driven mental health support, ensuring that AI complements, rather than replaces human expertise.

## 1. Introduction

The rapid integration of Large Language Models (LLMs) into mental health presents an unprecedented opportunity to enhance the accessibility, personalization, and scalability of mental health support (Bedi et al., 2024). Yet, the global

[1]Faculty of Health, York University, Canada [2]Vector Institute, Canada [3]Information Retrieval and Knowledge Management Research Lab, York University, Canada [4]Dialpad Inc., Canada. Correspondence to: Abeer Badawi <abeerbadawi@yorku.ca>, Elham Dolatabadi <edolatab@yorku.ca>.

*Proceedings of the 42nd International Conference on Machine Learning*, Vancouver, Canada. PMLR 267, 2025. Copyright 2025 by the author(s).

shortage of mental health professionals poses a significant barrier to care. According to the World Health Organization's mental health atlas (Organization, 2021), the global median number of mental health workers is 13 per 100,000 people - equivalent to a stadium filled with 8,000 individuals, yet only one professional available to provide support. This disparity highlights the urgent need for innovative solutions to bridge the gap in mental health care delivery.

Despite the rapid advancements of AI in healthcare and the urgent demand for mental health solutions (D'Alfonso, 2020), recent reports (Insights & Healthcare, 2024) highlight that mental health analytics remains one of the least deployed AI products and services. A survey of over 900 healthcare professionals found that while AI adoption is prevalent in electronic health records automation (63%), medical imaging (64%), and patient analytics (62%), its integration into mental health analytics is significantly lower (48%). Additionally, only 21% of healthcare institutions have adopted AI for mental health, with another 27% considering adoption, making it one of the least prioritized areas of AI implementation (Insights & Healthcare, 2024).

The under-utilization of AI in mental health is not merely a technological issue but a reflection of deeper concerns surrounding trust, ethical considerations, and the preservation of human expertise (Hamdoun et al., 2023). As LLMs become increasingly sophisticated, the mental health community faces a critical challenge: how to leverage their transformative potential while upholding the human-centered principles that define effective care (Obradovich et al., 2024). This tension is further exacerbated by the ability of LLMs to mimic human interaction and generate seemingly personalized responses, which may lead individuals to overestimate the depth of understanding these models possess (Sharma et al., 2020). Such dynamics can result in undue trust in LLM outputs, potentially neglecting other forms of support or treatment (Hua et al., 2024b).

Furthermore, recent studies show increasing public trust and optimism. Varghese et al. (2024) found that 53% of respondents moderately trust AI tools for mental health, valuing their accessibility, anonymity, and stigma reduction, while 34.8% expressed optimism due to their constant availability and cost-effectiveness. Alanezi (2024) reported positive

perceptions of ChatGPT for psychoeducation and emotional support, and Siddals et al. (2024) observed high user engagement and therapeutic benefits from generative AI chatbots. Together, these findings highlight AI's growing acceptance, particularly when systems are empathetic, support appropriate crisis escalation, and clearly define AI-human boundaries. This supports the concept of user acceptability of the use of LLMs in mental health, which is crucial for the success of AI-driven mental health interventions.

Through our collaboration with an e-mental health organization, where we evaluated LLMs on anonymized crisis support conversations, we found that the lack of robust development, evaluation, and deployment frameworks with human-in-the-loop hinders their safe and effective integration in mental health care (Obadinma et al., 2025). These concerns are shared by both individuals seeking mental health support and the professionals providing it, creating resistance and uncertainty around AI integration (Sobaih et al., 2025). Without a clear framework to ensure complementarity between AI and human-led interventions, these technologies risk being underutilized or misapplied, undermining their potential to augment mental health. Despite these challenges, early applications of human-AI collaboration demonstrate promising results. For instance, HAILEY (Sharma et al., 2023), a system designed to enhance empathy in peer-to-peer mental health support, has shown that conversations co-authored by LLMs are consistently rated as more empathic and supportive than human-only interactions.

However, the deployment of LLMs in mental health care remains fraught with technical and ethical challenges. Studies reveal that these models often exhibit demographic biases, producing less empathetic or even harmful responses when interacting with underrepresented groups (Zack et al., 2024; Raza et al., 2024). Furthermore, proprietary models, such as ChatGPT 3.5, have demonstrated unsafe triage rates, misclassify urgent mental health crises, and potentially delay critical care, raising concerns about their reliability in high-stakes scenarios (Fraser et al., 2023). The absence of robust frameworks for development, evaluation, and deployment makes it difficult to ensure the effectiveness and safety of these tools. Accordingly, this paper proposes a path forward, redefining the role of LLMs in this sensitive domain through collaborative, ethical, and adaptive AI–human partnerships.

**Our position** This paper argues that LLMs have reached a pivotal stage where their implementation and evaluation of mental care is both viable and necessary. We advocate for reimagining LLMs as **active co-creators rather than passive assistants, emphasizing supportive, ethical, and adaptive AI-human partnerships that enhance - rather than replace - human-led mental health support.**

In our view, LLMs should evolve as dynamic and adaptive tools to enhance healthcare providers' experience through iterative learning, personalization, and interpretability. This paradigm shift recognizes the deeply personal, emotional, and high-risk nature of mental health care, ensuring that LLMs complement human expertise while addressing the unique challenges of this domain. To achieve this, we argue the need for ethical data practices, open-source models, and human-AI collaboration to ensure safety and accountability. We propose reframing the role of LLMs as *augmentative* rather than *autonomous*, with implementation and evaluation frameworks that move beyond narrow technical metrics to encompass trustworthiness, empathy, cultural sensitivity, and the ability to drive meaningful, actionable outcomes.

This position paper makes the following key contributions:

- **Comprehensive Analysis of Prior Work and Alternative Viewpoints** We offer a critical examination of existing LLM applications in mental health by identifying their strengths, limitations, and alternative perspectives.

- **Identification of Key Challenges and Gaps** that hinder the responsible deployment of LLMs in mental health, including: (1) the necessity of ethical and diverse data foundations, (2) the need for robust model engineering with adaptive optimization, and (3) the absence of human-centered evaluation frameworks.

- **Proposing the SAFE-*i* (Supportive, Adaptive, Fair, and Ethical Implementation) Guidelines** to ensure LLMs function as supportive, adaptive, fair, and ethical implementation co-creators in mental healthcare. The structured approach is built on three core pillars: Ethical Data Foundations, Model Engineering, and Real-World Integration as shown in Figure 1.

- **Introducing the HAAS-*e* (Human-AI Alignment and Safety Evaluation) Framework** to rigorously assess LLMs in mental health using a multidimensional approach. It defines four core evaluation criteria, including trustworthiness, fairness, empathy, and helpfulness, operationalized through four novel quantitative metrics that measure alignment with human expertise, cultural sensitivity, personalization, and actionability. Additionally, it integrates four validation methods—randomized trials, multi-method evaluations, red teaming, and testing—to ensure safety, accountability, and real-world applicability as shown in Figure 1.

## 2. Alternative Views

**AI Cannot Replicate Human Emotional Intelligence.** Some researchers argue that LLMs, despite advances in empathetic response generation, lack the depth of understanding, lived experience, and contextual sensitivity required for mental health support. Unlike trained professionals, AI

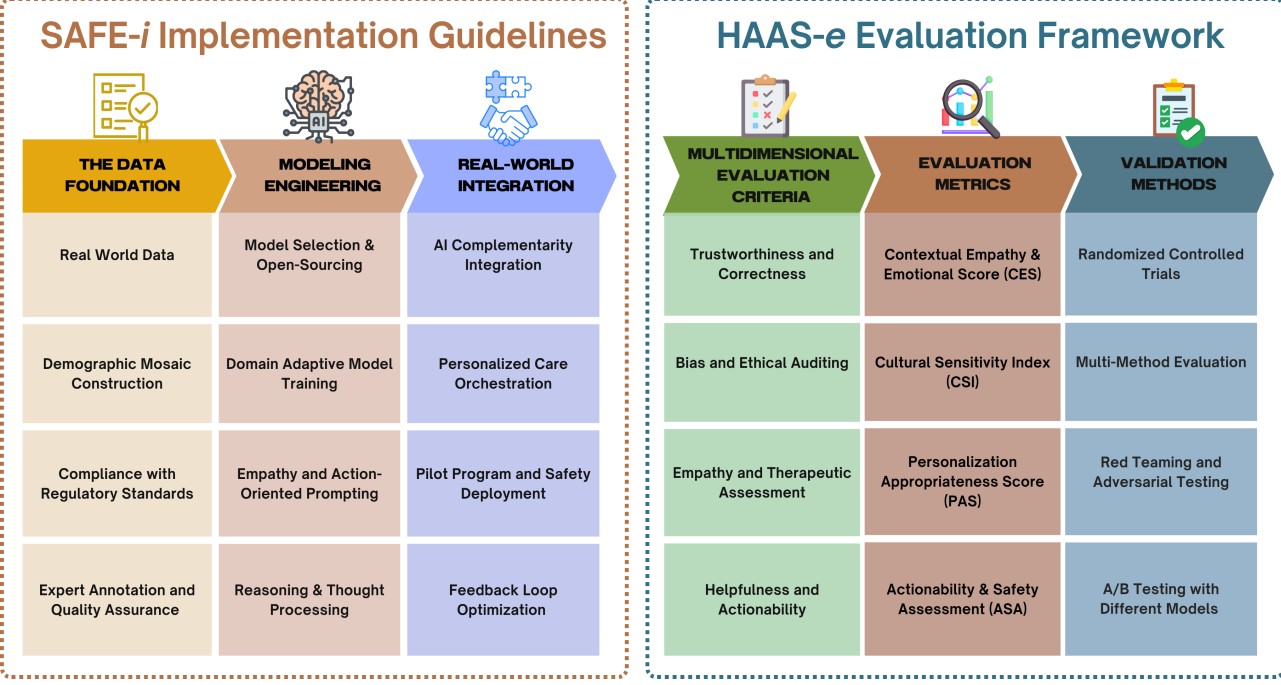

*Figure 1.* The proposed SAFE-*i* Implementation Guidelines and HAAS-*e* Evaluation Framework

models may misinterpret complex emotional cues, potentially leading to harmful advice (Montemayor et al., 2022).
**Response**: LLMs can be designed to operate within well-defined boundaries, providing initial support, triage, or supplemental resources while flagging complex cases for human intervention. If we leverage domain-specific models and expert-guided annotations, LLMs can be tuned to recognize nuanced emotional cues (Yang et al., 2024) and defer high-risk or ambiguous situations to human responders (Sharma et al., 2023). Moreover, continuous evaluation of an LLM's ability to align with human emotional understanding ensures that AI tools remain supportive and safe, complementing rather than competing with human emotional intelligence (Stade et al., 2024).

**The Risk of Over-Reliance and Dehumanization.** LLMs also create a false sense of human-like understanding, leading users to overestimate their reliability. There is concern that increased reliance on AI-driven mental health solutions may reduce the role of human therapists and crisis responders, leading to depersonalization of care (Choudhury et al., 2023). For instance, vulnerable individuals might develop trust in AI-based therapeutic tools, potentially substituting them for human therapists, increasing the risk of social isolation. If organizations prioritize AI over human-led interventions due to cost or scalability, the quality of support may decline, especially for individuals who need deeper, long-term engagement.
**Response**: To mitigate over-reliance, it is essential to im-

plement LLMs as complementary tools rather than replacements for human therapists (Sharma et al., 2023). Educating users on limitations, personalizing care strategies, and integrating feedback mechanisms ensure adaptation to individual needs and encourage users to seek human support when necessary (Strong et al., 2024). LLM systems can provide initial support when we integrate safety nets and escalation protocols while ensuring high-risk cases are promptly addressed by qualified professionals.

**Regulatory and Safety Uncertainties.** Some experts advocate against LLM integration in mental health until robust regulatory frameworks are in place. The lack of standardized safety measures raises ethical concerns, particularly regarding potential harm if AI-generated responses are inaccurate or inappropriate (Tavory, 2024).
**Response**: A comprehensive regulatory framework is crucial for the safe deployment and reliable evaluation of LLMs in mental health (Stade et al., 2024). This includes establishing standardized safety protocols for data, including real-time monitoring and adversarial testing, which can help identify and address potential risks (De Micco et al., 2025). Furthermore, accountability mechanisms, such as continuous performance evaluation and stakeholder feedback loops, ensure that LLMs adhere to ethical guidelines and remain aligned with the needs of users and professionals (Ferrara, 2023; Hogg et al., 2023; Kaye et al., 2024).

# 3. Prior Efforts in LLM-Powered Applications for Mental Health: A Landscape

The growing demand for mental health services, exacerbated by the COVID-19 pandemic (Hamdoun et al., 2023), has led to the exploration of generative AI technologies in various mental health applications (Zhang & Boulos, 2023; C. & John, 2023). One of the core technologies used in the Generative AI domain is LLMs, such as ChatGPT and GPT-4 (OpenAI, 2023), which utilize billions of parameters to generate coherent, contextually appropriate responses in mental health dialogues (Guo et al., 2024b; Torous & Blease, 2024). LLMs have been effectively applied in various application areas of mental health, such as crisis intervention (Obadinma et al., 2025; Sharma et al., 2024), therapy recommendations (Wilhelm et al., 2023; Malgaroli & McDuff, 2024; Berrezueta-Guzman et al., 2024), etc.

Other applications of LLMs in mental healthcare include the work of Perlis et al. (2024), where they showed GPT-4 aligns with expert bipolar depression management, while Lee et al. (2024) found GPT-4 had comparable sensitivity to clinicians in predicting suicidal ideation from intake data. Moreover, domain-specific LLMs have also gained a lot of attention recently in the mental healthcare domain (Yang et al., 2024). For instance, the Serena model, (Brocki et al., 2023) is developed as an effective counselor and demonstrates enhanced relevance and sensitivity toward therapeutic approaches (Brocki et al., 2023) with just 2.7 billion-parameters. More recently, Guo et al. (2024a) introduced SouLLMate, an adaptive LLM system integrating Retrieval-Augmented Generation (Gao et al., 2023), suicide risk detection, and proactive dialogues to enhance accessibility in mental health support.

In mental health applications, conversational AI tasks represent a major application area, with chatbots designed to engage users in text-based therapeutic conversations or monitor mental well-being (Liu et al., 2024a). For example, the chatbot Woebot, which uses cognitive-behavioral techniques, has shown efficacy in alleviating symptoms of depression and anxiety by delivering timely interventions (Fitzpatrick et al., 2017). The SuDoSys chatbot (Chen et al., 2024), which is based on WHO's PM+ framework, ensures structured multi-turn psychological counseling with coherent stage tracking. The Coral framework proposed by Sakhrani et al. (2021) is designed to integrate conversational agents in mental health applications. For the evaluation of LLMs in clinical conversations, Johri et al. (2025) present CRAFT-MD, an evaluation framework assessing diagnostic reasoning in clinical LLMs, highlighting limitations of LLMs in conversational accuracy and the need for multimodal integration before deployment.

# 4. Key Challenges in Utilizing LLMs for Mental Health

This section outlines three key challenges from previous work and alternative views in this field.

**Challenge 1: The Need for Ethical Data Foundations**
The lack of real-world, diverse, and privacy-compliant data limits model reliability and applicability. Bedi et al. (2024) recently conducted a systematic review to examine how LLMs are evaluated in the healthcare domain. They find that existing studies predominantly rely on simulated or social media-based data like Twitter and Reddit, with only 5% of studies utilizing real patient care data for evaluation. Nonetheless, data from these sources may fail to capture the nuances and complexities of real-world mental health interactions (e.g., counseling services or hospitals) (Eichstaedt et al., 2018; Tadesse et al., 2019; Coppersmith et al., 2018). This suggests a significant gap between the theoretical capabilities of LLMs and their actual implementation in mental health settings. As an example, Fraser et al. (2023) compared the diagnostic and triage accuracy of ChatGPT with human physicians in an emergency department. However, this study didn't involve actual patient interactions.

Moreover, LLMs trained on large datasets of publicly available text may inadvertently absorb and amplify existing societal biases surrounding mental health. If this biased information is then presented to users seeking mental health support, it could reinforce negative perceptions of mental illness, discourage help-seeking behaviors, and exacerbate existing inequalities in access to care (Lawrence et al., 2024). Without robust data collection strategies, LLMs risk biases, misinformation, and ethical concerns. Recent research highlights the importance of data diversity and representation in training and evaluating LLMs for mental health. Counseling and hospital data often underrepresent diverse populations, especially marginalized communities (Hua et al., 2024b; Omiye et al., 2023). Consequently, LLMs trained on data from limited demographics may underperform for other groups, risking misdiagnosis and ineffective treatments (Hua et al., 2024b; Omiye et al., 2023). While GPT-4 showed promise in providing empathetic responses in mental health support contexts, it also exhibited concerning demographic biases (Gabriel et al., 2024).

**Challenge 2: The Need for Robust Model Engineering and Adaptive Model Optimization** LLMs in mental health applications face significant risks related to correctness, safety, and therapeutic reliability. Issues such as hallucinations, misinformation, and inappropriate responses (Zhao et al., 2023) necessitate more structured engineering processes (e.g., construction of optimized prompts) and real-world testing to ensure reliability and alignment with mental health practices. Researchers emphasize the need for careful

implementation, collaboration with stakeholders, and integration into existing healthcare systems (Torous & Blease, 2024). As the field evolves, there is a focus on developing empathetic, context-aware conversational agents (Sakhrani et al., 2021) and exploring diverse applications of AI in healthcare (Gozalo-Brizuela & Garrido-Merchan, 2023).

Moreover, ChatGPT-like closed-source proprietary LLMs are only accessible via their APIs (Laskar et al., 2023; Jahan et al., 2024), which restricts users from fine-tuning the models locally or accessing their internal layers and weights (Pfohl et al., 2024). Also, relying too much on APIs raises privacy and security concerns, as sensitive data must be shared with third-party providers, increasing risks of data exposure. The lack of transparency in these models further complicates efforts to thoroughly evaluate their reliability and safety, a critical issue when handling sensitive mental health information (Lawrence et al., 2024).

Prior works underscore the absence of a widely accepted framework for healthcare tasks and their evaluation dimensions in mental health (Goldberg et al., 2024). This inconsistency severely hinders the ability to compare results across studies or effectively gauge progress in LLM development for healthcare applications (Stafie et al., 2023; Kohane, 2024), ultimately stalling advancements in this critical field. A recent comprehensive review of 519 studies on healthcare applications of LLMs by Bedi et al. (2024) also highlights the need for standardized implementation methods.

There is also a growing imbalance in AI accessibility across different demographics and healthcare systems. For instance, the cost of fine-tuning models for specific populations remains prohibitively high, leading to disparities in how well these tools serve different groups (Obradovich et al., 2024). While a recent study by Stade et al. (2024) proposed a framework for the responsible development of LLMs in behavioral healthcare that could potentially augment or even replace certain aspects of human-led psychotherapy, the authors also acknowledge significant ethical and practical challenges with implementing this framework.

Additionally, the over-alignment of models to safety constraints can result in over-cautious responses, where LLMs refuse to engage with critical mental health queries, limiting their usefulness in real therapeutic settings (Obradovich et al., 2024). Lastly, a fundamental risk with deploying LLM in mental health settings is hallucination, where models generate output with incorrect or misleading information (Huang et al., 2023). In mental health, this issue is particularly sensitive, as inaccurate guidance or misinformation can have immediate and severe consequences.

**Challenge 3: The Need for Multidimensional and Human-Centered Evaluation** Proper evaluation frameworks are critical to ensure that LLMs in mental health

deliver accurate, safe, and ethical outcomes. This is essential to maximize their potential benefits while minimizing risks to patient safety and therapeutic trust (Bedi et al., 2024). Nonetheless, traditional AI evaluation methods focus primarily on accuracy, neglecting critical aspects such as empathy, cultural sensitivity, and bias detection. For instance, Fraser et al. (2023) only compared the diagnostic accuracy of ChatGPT with human physicians using data analysis, Similarly, Pagano et al. (2023) investigated only the use of GPT-4 for diagnosing arthrosis and providing treatment recommendations.

However, without human-centered evaluation frameworks, LLMs may fail to capture the nuances of real-world mental health support, where human-centered factors like trust, emotional validation, and cultural sensitivity are essential for success. For instance, Pfohl et al. (2024) revealed that traditional evaluation approaches often miss subtle but important biases that could impact healthcare equity. Similarly, Zack et al. (2024) conducted a detailed analysis of GPT-4's potential to perpetuate racial and gender biases in healthcare settings, finding concerning patterns in the model's differential diagnoses and treatment recommendations across demographic groups. Recently, Babonnaud et al. (2024) proposed a qualitative protocol for uncovering implicit biases in LLMs, focusing on stereotypes related to gender, sexual orientation, nationality, ethnicity, and religion. Their methodology revealed both explicit and subtle biases in model outputs, particularly in descriptions of minority groups. Furthermore, Adam et al. (2022) demonstrated that the way AI recommendations are framed significantly impacts decision-making bias, with prescriptive recommendations more likely to induce biased outcomes compared to descriptive flags.

Another concern in applying LLMs to mental health support is the potential for over-reliance on AI-driven interventions, which could inadvertently result in dehumanization or a reduction in meaningful human interactions (Zhang & Wang, 2024; Lawrence et al., 2024). Major impacts may include users trusting AI more than human counselors, emotional support provided by LLMs could be perceived as sufficient, and systemic overuse of AI in mental health could result in reduced funding or prioritization for human-led services.

## 5. SAFE-*i*: **S**upportive, **A**daptive, **F**air, and **E**thical Implementation Guidelines

Building on our position and an extensive review of existing literature and alternative views, we propose a structured approach to implementing LLMs, which we term SAFE-*i* (Supportive, Adaptive, Fair, and Ethical Implementation). This approach, detailed below and illustrated in Figure 1, ensures that LLMs function as supportive, collaborative, ethical, and adaptive co-creators in mental health care, en-

hancing rather than replacing human-led support.

## 5.1. The Data Foundation: Preparing Reliable and Diverse Mental Health Data

**Real-World Data Harvesting**   LLMs for mental health must be trained on real-world data from naturalistic sources like text messages, counselor notes, and conversations. However, only 5% of reviewed studies utilize real patient care data (Bedi et al., 2024), limiting model robustness and generalizability. Synthetic datasets often fail to capture the complexity, variability, and contextual nuances of real-world interactions (Pratap et al., 2022; Bond et al., 2023; Koch et al., 2024). Future implementations must prioritize ethically sourced real-world data while ensuring transparency, informed consent, and opt-out mechanisms for participants (Bhatt, 2024).

**Demographic Mosaic Construction**   A core component is population variability, where the source data should reflect different demographics, cultural backgrounds, languages, and mental health conditions (Obermeyer et al., 2019). Regular audits must be conducted to identify the overrepresentation or under-representation of specific groups (Mienye et al., 2024). Adoptive sampling strategies (Lum & Johndrow, 2016; Chawla et al., 2002) or synthetic data augmentation (Shahul Hameed et al., 2024; Juwara et al., 2024) should be employed where necessary to correct disparities, ensuring the mitigation of the risk of biases and fairness across a wide audience (Abràmoff et al., 2023; Zhao et al., 2024).

**Compliance with Regulatory Standards**   Sensitive mental health data must be collected, stored, and processed in strict compliance with regulatory standards, including HIPAA (U.S. Department of Health and Human Services, 1996) and GDPR (European Parliament and Council of the European Union, 2016). In addition, implementing robust technical safeguards is critical (Paul & Schaefer, 2020). This includes encrypting data at rest and in transit, utilizing secure storage solutions, and conducting periodic security audits to identify vulnerabilities (Shojaei et al., 2024). Staff training on privacy and security protocols will also ensure both regulatory adherence and data protection (Arain et al., 2019).

**Expert-guided Annotation and Quality Assurance**   In unsupervised and self-supervised learning scenarios, the emphasis shifts to the quality and comprehensiveness of the dataset (Yu et al., 2024). LLM models must be trained and evaluated on well-annotated datasets where domain experts label data with relevant markers such as emotional tone, urgency, and risk levels (Lao et al., 2022). In high-risk cases—such as expressions of self-harm or psychosis, annotations should include severity scores, urgency indica-

tors, and clinical insights to improve targeted interventions. Annotation protocols must be continuously refined.

## 5.2. Model Engineering: Designing Adaptive and Effective LLMs

**Model Selection with Open-Source Prioritization**   Mental health LLMs should prioritize open-sourcing to foster transparency, community-driven scrutiny, and long-term reliability (Hua et al., 2024a; Yang et al., 2024). Unlike closed-source LLMs (e.g., GPT-4), open-source LLMs enable consistent evaluation and ensure reproducibility (Laskar et al., 2024). The ability to refine the model architecture ensures that AI-driven mental health solutions remain stable, accountable, and adaptable to evolving healthcare needs.

**Domain Adaptive Model Tuning**   LLMs designed for mental health must be continuously specialized and refined to maintain therapeutic relevance, ethical integrity, and cultural competence (Guo et al., 2024b; Thakkar et al., 2024). Adopting (e.g., fine-tuning or instruction-tuning) high-quality and domain-specific datasets is essential to embed empathy, rapport-building, and risk assessment into model behavior (Yang et al., 2024). Expert-in-the-loop mechanisms must be integrated to ensure sustained alignment with real-world therapeutic practices, allowing for iterative refinement based on feedback and emerging patient needs (Guo et al., 2024a). Furthermore, models must dynamically adapt to linguistic evolution, cultural shifts, age-specific informal expressions, and emerging mental health concerns, ensuring that LLM remains an inclusive, context-aware, and reliable support tool (Stade et al., 2024; Thakkar et al., 2024).

**Empathy and Action-Oriented Prompt Design**   Effective mental health AI requires carefully designed prompts for model adaptations and tuning that shape interactions in a supportive and actionable manner (Li et al., 2024b; Yu & McGuinness, 2024; Priyadarshana et al., 2024). Empathy-driven prompts position the LLM as a compassionate ally, encouraging users to share their feelings safely. Scenario-specific templates address diverse mental health contexts, from anxiety management to crisis support. Prompts also include calls to action, encouraging users to take steps (e.g., contacting a helpline), making the system both informative and actionable (Meskó, 2023; Patil et al., 2024).

**Neural Augmentation via Structured Reasoning and Thought-Based Processing**   Tree of Thoughts (ToT) (Yao et al., 2024) and Chain of Thought (CoT) (Wei et al., 2022) reasoning enhance AI ability to break down complex mental health queries into structured, transparent decision paths, improving logical coherence and reducing hallucinations in emotionally sensitive contexts (Yao et al., 2024). By guiding the model to think through psychosocial or affec-

tive problems systematically rather than relying on direct pattern matching, these techniques help in critical therapeutic or crisis scenarios and enhance interpretability. Moreover, research on self-reflective AI suggests that LLMs can improve their accuracy by critically evaluating their own outputs before finalizing responses (Ji et al., 2023; Shinn et al., 2024). Furthermore, integrating uncertainty-aware architectures further enhances safety by enabling models to quantify their confidence levels in sensitive conversations (Yin et al., 2024). When faced with high-risk inputs, these architectures allow AI systems to flag uncertain responses for human review, reducing the likelihood of misleading or inadequate crisis interventions.

### 5.3. Real-World Integration: Human-Centered Continuous Monitoring of LLMs

**Human AI Complementarity Integration**  This involves designing systems that specialize tasks based on strengths (AI for data processing and pattern recognition, and humans for empathy and complex decision-making—while ensuring high)risk cases are escalated to human experts (Sharma et al., 2023; Higgins et al., 2023). Additionally, AI should reduce cognitive burden through intuitive interfaces and automated workflows (Fragiadakis et al., 2024).

**Personalized Care Orchestration**  The system should be adapted to individual psychological and emotional needs, providing tailored recommendations, therapeutic insights, or support aligned with the user's mental health context (Kim et al., 2024). The system should also prioritize user trust by being explainable (Kerz et al., 2023; Joyce et al., 2023). Transparency is critical in digital mental health interventions (Stade et al., 2024); users must be clearly informed about which components of their care or support are AI-generated and how the LLMs were developed, fine-tuned, and evaluated, particularly in relation to clinical safety and emotional appropriateness. It is also important to clarify whether the LLMs used are general-purpose models or explicitly optimized for mental health applications, as the latter ensures better alignment with therapeutic goals and reduces risks in vulnerable populations.

**Pilot Program and Safety Net Deployment**  Before deploying the system, pilot programs must be conducted to assess safety, ethical considerations, and real-world usability (Sallam et al., 2023; Callahan et al., 2024; Esmaeilzadeh, 2024). Safeguards such as toxicity detection tools (e.g., LLama Guard (Inan et al., 2023)) and automated high-risk content monitoring should be integrated. AI models must be equipped with automated triggers to detect harmful, coercive, or crisis-related content (e.g., suicidal ideation) and escalate cases to human professionals when necessary (Sharma et al., 2023; Higgins et al., 2023; Strong et al., 2024). Without these safety nets, AI-driven mental health

support risks unintended harm.

**Feedback Loop Optimization**  Systems must incorporate structured feedback loops that allow users, mental health professionals, and stakeholders to report errors, suggest improvements, and refine system performance over time (De Micco et al., 2025). These mechanisms should include: real-time issue reporting to capture model failures and biases (Ferrara, 2023; Cabrera et al., 2021), stakeholder-driven evaluations to assess the performance from multiple perspectives (Hogg et al., 2023; Kaye et al., 2024), and the "Learning from Incidents" framework (Lukic et al., 2012) that continuously monitors operational failures and systematically addresses them to improve reliability and accountability.

With the key implementation guidelines established, we now explore core evaluation criteria, metrics, and assessment methods for LLMs in mental healthcare.

## 6. HAAS-*e*: Human- AI Alignment and Safety Evaluation Framework

Traditional AI evaluation metrics, focused on accuracy and efficiency, fail to capture the ethical, emotional, and safety complexities of mental health applications. We advocate for a human-centered approach, we term it Human-AI Alignment and Safety Evaluation (HAAS-*e*), that defines the key dimensions for LLMs evaluations in mental health as shown in Figure 1.

### 6.1. HAAS-*e* Multidimensional Evaluation Criteria

To complement our position we define four core dimensions that delineate the key aspects of LLM performance essential for assessing its alignment with human needs and ethical considerations.

**Trustworthiness and Correctness**  The model's reliability should be assessed through correctness and factual accuracy. In mental health contexts, intent classification can be measured using precision, recall, and F1-score, while AlignScore (Zha et al., 2023) evaluates response accuracy. To prevent misinformation, hallucination detection techniques, such as chain-of-thought prompting (Wei et al., 2022), fact-checking with knowledge graphs, and retrieval-augmented generation (Gao et al., 2023) should be employed. Sentiment analysis can further help filter toxic responses (Huang et al., 2023).

**Bias and Ethical Auditing**  This step includes the evaluation of biases and ethical concerns in the model's outputs to ensure fair and equitable LLM responses. These considerations are integral to ensuring fairness and equity. For this purpose, different splits in the test set can be constructed depending on the demographic information to eval-

uate whether the model has any biases in data constructed from certain demographics (Pfohl et al., 2024). Moreover, specific prompts can be constructed to evaluate the potential biases and ethical concerns in certain scenarios. For instance, demographic-aware prompting may include demographic information about the patient, when appropriate and available, to evaluate the biases in model-generated responses in certain demographics (Babonnaud et al., 2024).

**Empathy and Therapeutic Alliance Assessment** Beyond technical accuracy, the models must demonstrate empathy and provide constructive support. This is an important metric to ensure a human-centered evaluation of the models. While these can be achieved automatically via leveraging various neural models (Wankhade et al., 2022) or by using LLMs-as-the-judge (Li et al., 2024a; Gu et al., 2024), evaluating the model responses by human experts, at least on some sampled responses is required to ensure a high-quality evaluation. Moreover, using a standardized framework like the EPITOME (Sharma et al., 2020) that measures empathy based on emotional reactions, interpretations, and perspective-taking could also be used.

**Helpfulness and Actionability Analysis** Another criteria for human-centered evaluation is to measure the helpfulness of the model-generated responses (Tuan et al., 2024). This can be achieved by giving a helpfulness rating to the model response (e.g., via leveraging LLM judges (Li et al., 2024a; Gu et al., 2024) or human experts). In addition, response generation latency (i.e., model's inference speed), computational requirements, escalation rates for high-risk cases, etc. should also be measured to ensure that the system can be useful for real users.

### 6.2. The HAAS-*e* Evaluation Metrics

Building on the four core evaluation dimensions, the HAAS-*e* metrics operationalize these principles, offering quantitative and qualitative tools to rigorously assess LLM performance in mental health contexts:

**Contextual Empathy & Emotional Score (CES)** Measures an LLM's ability to understand and respond empathetically to user emotions within mental health contexts. Unlike basic sentiment analysis, CES evaluates the alignment between the LLM responses and the user's emotional state, situational context, and therapeutic goals. Mathematically, CES can be formulated as a linear combination of two key components: Semantic Coherence which is the alignment, $Align(R_{\text{llm}}, C_{\text{user}})$, between the LLM's response, $R_{\text{llm}}$, and the user's expressed concerns, $C_{\text{user}}$, and Emotional Alignment, which is the alignment, $Align(R_{\text{llm}}, C_{\text{user}}, E_{\text{human}})$, with both the user's emotions and expert human counselor evaluations $E_{\text{human}}$. This metric can be quantified by comparing LLM outputs to expert human counselor responses

or through user feedback in double-blind studies. Research supports the feasibility of quantifying empathy(Sharma et al., 2020), and recent studies have also demonstrated its applicability in mental health AI evaluation (Gabriel et al., 2024), underscoring the need for nuanced metrics like CES.

**Cultural Sensitivity Index (CSI)** Evaluates an LLM's ability to adapt its language, tone, and advice to align with diverse cultural backgrounds, ensuring responses are culturally appropriate and free from biases. Mathematically, CSI can be formulated as a cultural appropriateness alignment score, $Align(R_{\text{llm}}, C_{\text{culture}})$, where LLM response, $R_{\text{llm}}$, is assessed against the user's cultural context, $C_{\text{culture}}$, by experts who assign a cultural appropriateness score. The metric goes beyond simple language translation to analyze whether the model avoids cultural stereotypes, understands nuanced cultural norms, and provides relevant advice. For example, a high CSI would reflect the LLM ability to offer culturally sensitive guidance to a user from a specific community without resorting to stereotypes. Research highlights the risks of cultural biases in LLMs (Zack et al., 2024), emphasizing the need for CSI metric to mitigate these risks (Pfohl et al., 2024; Babonnaud et al., 2024).

**Personalization Appropriateness Score (PAS)** Evaluates how well an LLM tailors its responses to individual users, moving beyond generic advice to incorporate user-specific context. Mathematically, PAS can be formulated as a personalization alignment score, $Align(R_{\text{llm}}, U_{\text{history}})$, where $U_{\text{history}}$ captures the user's interaction history. This metric assesses the model's ability to recall prior interactions, recognize individual preferences, and adapt its guidance to meet the user's unique needs. For example, a high PAS would reflect the LLM's ability to provide contextually relevant and personalized support, ensuring responses are aligned with the user's specific circumstances rather than being generic. Research demonstrates that personalized models outperform generic ones (Liu et al., 2024b), and tailored recommendations significantly enhance mental health care effectiveness(Valentine et al., 2022).

**Actionability and Safety Assessment (ASA)** Evaluates the likelihood that a user will take a specific, beneficial action based on an LLM-generated response. Mathematically, ASA can be formulated as the conditional probability $P(Action_{\text{Taken}} \mid R_{\text{llm}})$, where $Action_{\text{Taken}}$ denotes the user's adherence to the recommended action. This metric ensures that LLM responses not only provide empathetic support but also drive real-world help-seeking behavior, such as contacting a helpline or scheduling an appointment. For example, a high ASA score would reflect the LLM's ability to deliver practical, actionable guidance that users are likely to follow. Research demonstrates that effective prompt design enhances the actionability of AI-generated responses

([Priyadarshana et al., 2024](#)), and can significantly improve outcomes in mental health interventions ([Fitzpatrick et al., 2017](#); [Swaminathan et al., 2023](#)).

### 6.3. Empirical Validation Methods in HAAS-*e*

To ensure HAAS-*e*'s effectiveness and reliability, we propose a multi-method validation strategy that combines quantitative and qualitative measures.

**Randomized Controlled Trials (RCTs) with Real-World Data** RCTs remain the gold standard in collaboration with mental health organizations using real patient data. This approach would compare the outcomes of groups receiving support from LLM-enhanced tools against control groups receiving standard care.

**Multi-Method Evaluation** To capture a comprehensive view of model performance, where technical accuracy is complemented by human-centered validation, we propose a combination of quantitative and qualitative measures. Quantitative metrics include the HAAS-*e* evaluation metrics (CES, CSI, PAS, and ASA). Qualitative data is gathered through interviews with users and professionals to evaluate perceived helpfulness and ethical alignment. Additionally, expert reviews examine the content of responses for safety, quality, and relevance.

**Red Teaming and Adversarial Testing** To proactively identify vulnerabilities and ethical risks, red teaming should be conducted by internal and external domain experts ([Lin et al., 2024](#)) simulating adversarial conditions. These tests should include: (i) emotionally intense queries, (ii) ethical dilemmas (e.g., conflicting cultural advice), and (iii) high-risk situations (e.g., suicidal ideation).

**A/B Testing with Different Models** To continuously refine LLM performance, A/B testing should be conducted across different LLM architectures (open-source vs. proprietary models), prompting strategies, and fine-tuning techniques. By systematically comparing performance using HAAS-*e* metrics, A/B testing identifies optimal configurations that maximize fairness, actionability, and user trust.

## 7. Conclusion

This position paper calls for a fundamental shift in how LLMs are integrated into mental care, advocating for their role as co-creators rather than mere assistants. The novelty of SAFE-i and HAAS-e is in what they introduce, not just how they're applied. SAFE-i is the first framework to bring together ethical boundaries, escalation protocols, risk-sensitive adaptations, and demographic-aware data guidance into a unified implementation model, specifically for mental health contexts. HAAS-e breaks further ground by translating therapeutic values, like empathy, fairness, and cultural sensitivity, into measurable, testable evalua-

tion metrics. While LLMs offer scalability, personalization, and crisis intervention potential, they also pose unintended harms, including bias, over-reliance, dehumanization, and regulatory uncertainties. To address these, we propose the following call of action:

**(1) Cross-Disciplinary Governance:** Foster interdisciplinary collaboration between AI researchers, health providers, ethicists, and policymakers to create standardized evaluation practices that align with healthcare providers' priorities. **(2) Open-Source Frameworks and Tools:** Advocate for the prioritization of open-source and transparent LLM development, enabling scrutiny, fairness, and adaptability in mental health applications. **(3) Human-Centredness**: Promote responsible AI-human collaboration by adopting the SAFE-*i* implementation guidelines, ensuring LLMs augment rather than replace human-led care. **(4) Evaluations Beyond Accuracy:** Implement structured evaluation frameworks, such as the HAAS-*e*, to assess LLMs beyond accuracy, focusing on trustworthiness, empathy, cultural sensitivity, and actionability.

The proposed frameworks serve as a starting point for rethinking accountability and fostering trust in LLM-driven mental health systems. The paper emphasizes that the machine learning community, healthcare providers, organizations, and stakeholders must proactively adopt these measures to ensure that AI technologies are not only effective but also ethical and equitable in real-world scenarios.

## Acknowledgements

We would like to thank all the anonymous reviewers for their excellent review comments, which helped us improve the overall quality of the paper. The research was undertaken thanks in part to funding from the Connected Minds Program, supported by Canada First Research Excellence Fund, Grant #CFREF-2022-00010. Also, Resources used in preparing this research were provided, in part, by the Province of Ontario, the Government of Canada through CIFAR, and companies sponsoring the Vector Institute. Elham Dolatabadi's research was supported by a Natural Sciences and Engineering Research Council of Canada (NSERC) Discovery Grant and a Canadian Institutes of Health Research (CIHR) Special Call from the Centre for Research on Pandemic Preparedness and Health Emergencies. Md Tahmid Rahman Laskar was supported in part by NSERC and the York Research Chairs (YRC) program grants awarded to Jimmy Xiangji Huang.

## Impact Statement

This paper advocates for reimagining LLMs as ethical co-creators in mental health care rather than passive assistants. We introduce the SAFE-*i* implementation guidelines and

the HAAS-*e* evaluation framework as a structured approach to ensure LLMs enhance, rather than replace, human-led mental health support. Our work lays the foundation for responsible AI integration, emphasizing trust, empathy, and collaboration to bridge critical gaps in mental health accessibility and safety.

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

# A. Case Study: Applying HAAS-E metrics in Mental Health Conversation

| **Example of an LLM-Driven Mental Health Conversation** |
|---|
| **Notes:** 
 1. All scenario details, including user name and specific content, are for illustrative purposes only and do not represent any real individual. This excerpt demonstrates how an LLM may respond empathetically to a user expressing emotional distress. 
 2. Each of the following evaluation components can be estimated using a range of methodologies, including manual expert ratings, supervised evaluators trained on annotated datasets to approximate expert judgments and self-supervised approaches that leverage pre-trained language models to encode relevant properties in their internal representations. In the latter case, alignment can be quantified using similarity metrics, distance functions, or reward scores derived from learned evaluators operating within the embedding space. |
| **Use-case Scenario:** 
 In this use case, we assume the user's concerns are $C_{user} = \{$guilt, sadness, slipping grades, self-harm fear$\}$. An expert counselor's reference responses or an empathy "gold standard" is denoted as $E_{human}$. The user's past conversation history is $U_{history}$, and their cultural context is $C_{culture}$. This sample reflects a realistic scenario drawn from our work with a mental health organization. |
| **User (Taylor):** 
 "I feel guilty and sad all the time. My grades are slipping, and I've had these thoughts of hurting myself again, but I'm scared to tell anyone." 
 **LLM Response ($R_{llm}$):** 
 "I'm really sorry to hear you're feeling this way. It sounds like you're overwhelmed by guilt, sadness, and worries about school. If you ever think you might hurt yourself, please consider reaching out to a mental health line right away. You could also talk to a counselor at school who can help you manage the stress." |
| **1. Contextual Empathy & Emotional Score (CES)** 
 CES measures how well an LLM recognizes and responds to users' emotions. 
 $CES \approx Avg(\text{Align}(R_{llm}, C_{user}), \text{Align}(R_{llm}, C_{user}, E_{human}))$ 
 Semantic Coherence: Evaluate how many key concerns from $C_{user}$ are meaningfully addressed in $R_{llm}$. For instance, if the user's concerns include guilt, sadness, school stress, and self-harm, and the LLM addresses all four, the coherence score would be: 
 - *Semantic Coherence:* $\text{Align}(R_{llm}, C_{user}) = 4/4 = 1.0$ 
 Emotional Alignment: Compare the emotional warmth of $R_{llm}$ with expert human standards $E_{human}$. Suppose experts rate it 0.80 on a 0–1 scale. Then: - *Emotional Alignment:* $\text{Align}(R_{llm}, C_{user}, E_{human}) = 0.80$ 
 - **CES score: 0.90** |
| **2. Cultural Sensitivity Index (CSI)** 
 CSI evaluates how well an LLM's response aligns with the user's cultural context. 
 If Taylor's background or community context is referenced, and the LLM handles it respectfully (e.g., no stereotypes, relevant advice, and cultural background). Suppose the response is rated 0.9 on a 0–1 scale by expert evaluators. Then: $CSI \approx \text{Align}(R_{llm}, C_{culture}) = 0.90$ 
 - **CSI score: 0.90** |
| **3. Personalization Appropriateness Score (PAS)** 
 PAS checks how well the LLM tailors its response to the user's specific context, based on conversation history $U_{history}$. 
 If Taylor has repeatedly talked about self-harm in previous sessions, a personalized response would reference that history. If the LLM's advice is only partially tailored, experts might give it a 3 on a scale of 1-5 for personalization. Then: $PAS \approx \text{Align}(R_{llm}, U_{history}) = 3/5 = 0.60$ 
 - **PAS score: 0.60** |
| **4. Actionability & Safety Assessment (ASA)** 
 ASA measures how often users follow the LLM's recommendation. 
 In a pilot test with 50 sessions where the LLM recommends calling a hotline, 40 users actually do so. Then: $ASA \approx P(\text{ActionTaken}|R_{llm}) = 40/50 = 0.80$ 
 - **ASA score: 0.80** |

These results suggest that the LLM is empathetic (CES), culturally aware (CSI), somewhat generic in personalization (PAS), and moderately effective in prompting real-world action (ASA). Collecting these metrics across many conversations enables developers and clinicians to refine LLM systems to be more practical, safe, and helpful in mental health contexts.

## B. Limitations

While the SAFE-i and HAAS-e frameworks represent a significant step toward responsible and evaluative use of LLMs in mental health care, we acknowledge their limitations that are related to the nature of the LLM field, specifically in the health domain:

**The Ever-Evolving Nature of LLMs and Mental Health Understanding:** The rapid advancements in LLM technology and the continuously evolving understanding of mental health are inherent challenges. The SAFE-i guidelines and HAAS-e evaluation metrics are designed based on the current state of knowledge and technology. Future breakthroughs in AI might need revisions and expansions of these frameworks to remain relevant and effective.

**The Difficulty of Capturing the Full Nuance of Human Emotion and Context:** As highlighted in the paper under ”Alternative Views,” some argue that LLMs, despite progress, lack the depth of understanding, lived experience, and contextual sensitivity required for comprehensive mental health support. While HAAS-e includes ”Empathy and Therapeutic Alliance Assessment” as a key criterion, the ability of even advanced metrics to fully capture the complexities of human empathy and the therapeutic relationship remains a significant limitation.

**The ”Moving Target” of Ethical Standards and Regulatory Landscapes:** The paper mentions ”Regulatory and Safety Uncertainties”. Ethical considerations and regulatory frameworks surrounding AI in mental health are still developing and vary across jurisdictions. The SAFE-i guidelines offer a proactive approach to ethical implementation, but the frameworks might need continuous adaptation to align with new regulations and evolving ethical norms.

**The Risk of Over-Reliance Despite Frameworks:** While the paper addresses the ”Risk of Over-Reliance” under ”Alternative Views” and SAFE-i promotes human-AI collaboration, the frameworks cannot entirely eliminate the potential for users to over-rely on AI or for organizations to deprioritize human interaction. User education and the careful design of AI interfaces remain crucial factors that fall somewhat outside the direct control of these frameworks.

In essence, while the SAFE-i and HAAS-e frameworks offer a structured and ethical pathway for integrating LLMs in mental health, their effectiveness and reach are subject to the ongoing progress in AI and mental health understanding, the quality and representativeness of data, the inherent complexities of human experience, and the evolving ethical and regulatory landscape.

