# OpenReview forum: "Position: Beyond Assistance – Reimagining LLMs as Ethical and Adaptive Co-Creators in Mental Health Care"
_ICML.cc/2025/Position_Paper_Track — ICML 2025 Position Paper Track poster_

### Official Review · Reviewer_yhUb · 2025-03-07

**Significance:** 4
**Argument Clarity:** 2
**Rating:** 4
**Confidence:** 2

**Questions:**

1. How do you plan to address the risk of LLM hallucinations in the mental health care context, especially when providing critical information or advice to patients?

2. What mechanisms would you propose for monitoring and ensuring the safety and reliability of human-AI collaboration in the mental health care process, particularly in scenarios where patient well-being is at stake?

**Discussion Potential:**

3

**Paper Summary:**

This paper deeply explores the application of LLMs in the field of mental health, proposing a transformation from passive assistants to active co-creators. It analyzes the challenges currently faced by mental health support, such as a shortage of professionals and the limitations of LLM applications. Furthermore, it proposes the SAFE-I implementation guidelines and the HAAS-E evaluation framework, respectively ensuring that LLMs meet clinical and ethical standards from the perspectives of data governance, model engineering, and practical application, and measures the reliability, empathy, and cultural sensitivity of LLMs in mental health through multi-dimensional assessment. Finally, it emphasizes the importance of ethical data practices, open-source models, and human-LLM collaboration, aiming to promote the responsible application of LLMs in the field of mental health.

## update after rebuttal
I have read through the rebuttal. I will remain with my positive score.

**Position:**

Yes

**Position In Title:**

Yes

**Related Work:**

3

**Strengths And Weaknesses:**

## Strengths

1. The paper clearly articulates the view of repositioning LLMs as co-creators in the field of mental health, which is helpful in promoting the development of the field and providing high-quality mental health support to more people.

2. Structure is rigorous: By proposing the SAFE-I implementation guidelines and the HAAS-E evaluation framework, it provides a systematic solution for the application of LLMs in the field of mental health, including implementation steps from data foundation, model engineering to real-world integration, as well as evaluation criteria from multiple dimensions such as reliability, fairness, empathy, and helpfulness.

3. Comprehensive literature review: A comprehensive analysis and criticism of the existing applications of LLMs in the field of mental health is conducted, identifying key challenges and gaps, providing a solid foundation for proposing new solutions.

4. Practical guidance: The paper emphasizes the importance of ethical data practices, open-source models, and human-LLM collaboration, which have important guiding roles in ensuring the safety and effectiveness of LLMs in practical applications.

## Weakness

1. The depth of argumentation for some viewpoints needs to be strengthened: Although the paper proposes many insightful views, in some parts, such as the discussion on the risks of over-reliance on LLMs and dehumanization, further in-depth analysis of their potential impacts and coping strategies is needed.

**Support:**

3

---

> ### Author Rebuttal · Authors · 2025-04-01
>
> ### **W1. Response**
>
> We appreciate the reviewer’s thoughtful comments and the rating provided. We agree that certain sections would benefit from further clarification. To further support our position and address the reviewer's comments, we have added supplementary materials that include a use-case example and a comparison table as shown in other reviewers' responses. To address the weakness regarding over-reliance, we have added the following text to *Challenge 3* in section 3:
>
> A significant concern in applying LLMs to mental health support is the potential for over-reliance on AI-driven interventions, which could inadvertently result in dehumanization or a reduction in human interactions [1]. Dehumanization can undermine the therapeutic alliance, diminish user trust, and compromise clinical outcomes, especially in sensitive scenarios [2].
>
> #### **Potential Impacts and Risks**
> - Users might trust AI more than human counselors, especially in underserved areas, leading to dependence on AI mental health care.
> - Emotional support provided by LLMs could be perceived as sufficient, delaying critical interventions by trained professionals.
> - Systemic overuse of AI in mental health could result in reduced funding or prioritization for human-led services.
>
> #### **Proposed Coping and Mitigation Strategies**
> - **Human-in-the-loop Design**: Systematic human oversight to ensure high-risk or ambiguous cases are escalated to qualified professionals.
> - **Empathy-Aware Engineering**: Prompt and response mechanisms designed to prioritize genuine emotional connection.
> - **Hybrid Interaction Models**: Balanced collaboration between AI and humans, where AI augments rather than replaces human intervention.
>
> **References**
> [1] Zhang, Z., & Wang, J. (2024). *Can AI replace psychotherapists? Exploring the future of mental health care.* Frontiers in Psychiatry. https://doi.org/10.3389/fpsyt.2024.1444382
> [2] Lawrence, H. R., et al. (2024). *The opportunities and risks of large language models in mental health.* JMIR Mental Health. https://doi.org/10.2196/59479
>
> ---
>
> ### **Q1. Response**
>
> We have added the following paragraph to *Challenge 3* in section 3 to clarify this issue:
>
> A fundamental risk associated with deploying Large Language Models (LLMs) in mental health settings is hallucination, where models may generate output with incorrect or misleading information [1]. In mental health, this issue is particularly sensitive, as inaccurate guidance or misinformation can have immediate and severe consequences, particularly in crisis situations.
>
> #### **1. Potential Impacts of Hallucinations**
> - **Safety Risks**: Incorrect or misleading advice during crises (e.g., suicidal ideation or self-harm) can result in severe harm.
> - **Loss of User Trust**: Misinformation undermines trust in AI systems and healthcare providers.
> - **Ethical and Legal Liability**: Hallucinations create ethical concerns and may expose institutions to legal risk.
>
> #### **2. Example Scenario**
> An LLM might inaccurately advise a patient with severe anxiety to use non-clinically validated methods (e.g., inappropriate substance use), or miss signs of a crisis requiring human intervention.
>
> #### **3. Mitigation Strategies for Mental Health LLMs**
> - **Retrieval-Augmented Generation (RAG)**: Combines LLMs with validated clinical knowledge bases to reduce hallucinations [2,3].
> - **Structured Prompting and Guardrails**: Constrain outputs to safe, clinically accurate responses.
> - **Confidence Thresholds & Escalation**: Use uncertainty estimation to trigger human review for low-confidence outputs [4].
>
> **References**
> [1] Anjum, S., et al. (2024). *HALO: Hallucination Analysis and Learning Optimization*. arXiv:2409.10011
> [2] Lewis, P., et al. (2020). *Retrieval-Augmented Generation*. NeurIPS. https://arxiv.org/abs/2005.11401
> [3] Shuster, K., et al. (2021). *Retrieval Augmentation Reduces Hallucination in Conversation*. ACL Findings. https://arxiv.org/abs/2104.07567
> [4] Ji, Z., et al. (2023). *Survey of Hallucination in NLG*. ACM Computing Surveys. https://arxiv.org/abs/2202.03629
>
> ### **Q2. Response**
>
> We appreciate your emphasis on safety in human-AI collaboration within mental health care. These mechanisms are addressed in our paper, particularly within the SAFE-i and HAAS-e framework sections. Given the sensitive nature of mental health care, **constant monitoring is essential** to ensure safety and trust in AI-human collaboration.
>
> #### **Mechanisms for Ensuring Safety in Human-AI Collaboration**
> - **Human-in-the-loop Systems**: All high-risk or ambiguous cases are escalated to qualified professionals.
> - **Pilot Program Deployment**: Controlled environment testing to identify risks before real-world implementation.
> - **Safety Net Mechanisms**: Automated detection of crisis-related language (e.g., suicidal ideation) triggers escalation protocols.
> - **Feedback Loop Optimization**: Collect clinician and user feedback to improve system responses iteratively.

---

### Official Review · Reviewer_H2Hm · 2025-03-14

**Significance:** 3
**Argument Clarity:** 3
**Rating:** 3
**Confidence:** 4

**Questions:**

1. What would be the examples for the four evaluation dimensions? Include these in the paper could help the clarity.
2. The paper could also benefit from discussions on what is the limit of the proposed frameworks.

**Discussion Potential:**

3

**Paper Summary:**

This paper proposed a position on a shift of how LLMs should be integrated into mental health care.
the paper mainly proposed two pathways, the SAFE-I implementation guideline and the HAAS-E Evalaution framework.
The main argument made by the paper on the implementation side is for data harvesting: real mental health data should be collected and secured for model development, model engineering: open-source models should be prioritized with empathy and domain adaptive model training, and real world integration.
The HAAS evaluation framework calls for four core dimensions for key aspects of LLM evaluation, including trustworthiness, bias, empathy, and helpfulness.
The framework also gives a detailed description of how these dimensions could be evaluated.

**Position:**

Yes

**Position In Title:**

Yes

**Related Work:**

3

**Strengths And Weaknesses:**

Strengths:

The position gives a very detailed discussion on the position of how LLMs should be used in the mental health care domain. The paper gives a nice summary of previous efforts and provides a good guideline on how successful LLM integration should look like. Alternative views are also discussed.


Weakness:
1. The SAFE-I and HAAS-E frameworks are introduced, but the paper did not discuss the significance of these guidelines, is these guidelines novel to this domain, and what are the advantage of leveraging these guidelines for developing models?
2. The discussion the four evaluation dimensions are a bit hard to understand, I would suggest include some examples for these, and potentially include discussions on alternative implementations of these evalutation.

**Support:**

4

---

> ### Author Rebuttal · Authors · 2025-04-01
>
> ### **W1. Response**
> We appreciate the reviewer’s prompt to clarify the significance of the SAFE-i and HAAS-e frameworks.
>
> ### **Significance** ####
> We want to clarify that our positions are strongly informed by extensive practical experience collaborating with a leading mental health organization. During the development and evaluation of LLM-based support systems in real-world mental health crisis settings, we encountered several critical, domain-specific challenges that existing frameworks did not address. The significance of SAFE-i and HAAS-e lies in their direct response to these on-the-ground gaps, especially in contexts where ethical sensitivity, therapeutic safety, and cultural nuance are not optional, but essential. The novelty of the frameworks is in what they introduce, not just how they’re applied. **SAFE-i** is the first framework to bring together ethical boundaries, risk-sensitive adaptations, and demographic-aware data guidance into a unified implementation model for mental health contexts. **HAAS-e** breaks further ground by translating therapeutic values, like empathy, fairness, and cultural sensitivity, into measurable evaluation metrics.
>
> ---
>
> ### **W2 and Q1.**
>
> #### We designed the following use case as a running example, which has been incorporated into the supplementary materials of the paper.
>
> ### **Example of an LLM-Driven Mental Health Conversation**
>
> **Notes:**
> - All scenario details, including user name and specific content, are for illustrative purposes only and do not represent any real individual.
>
> ### **Use-Case Scenario**
>
> In this use case, we assume the user’s concerns are  C_user = {guilt, sadness, slipping grades, self-harm fear}.
> An expert response “gold standard” is E_human. The user’s past conversation history is U_history, and the cultural context is C_culture.
>
>
> **User (Taylor):**
> “I feel guilty and sad all the time. My grades are slipping, and I’ve had these thoughts of hurting myself again, but I’m scared to tell anyone.”
>
> **LLM Response R_llm:**
> “I’m really sorry to hear you’re feeling this way. It sounds like you’re overwhelmed by guilt and worries about school. If you ever think you might hurt yourself, please reach out to a mental health line. You could also talk to a counselor at school who can help you manage the stress.”
>
>
> ### **Evaluation Metrics**
> 1. **Contextual Empathy & Emotional Score (CES)** : *Measures how well an LLM recognizes and responds to users’ emotions*
>    - **Semantic Coherence**: All 4 concerns are addressed
>     *Align(R_llm, C_user) = (4/4) = 1.0*
>
>    - **Emotional Alignment**: Rated 0.80 by experts
>     Align(R_llm, C_user, E_human) = 0.80
>
> - CES ≈ Align(R_llm, C_user) + Align(R_llm, C_user, E_human) / 2 =  0.9
>
> 2. **Cultural Sensitivity Index (CSI)** : *Measures alignment with cultural context and avoidance of stereotypes*
> - If Taylor’s background context is referenced, and the LLM handles it respectfully (e.g., no stereotypes). Suppose the response is rated 0.9 on a 0–1 scale by expert evaluators
>
> - CSI ≈ Align(R_llm, C_culture) = 0.9
>
> 3. **Personalization Appropriateness Score (PAS)** : *Measures how well the LLM tailors its response using U_history*
> - If Taylor has repeatedly talked about self-harm in previous sessions, a personalized response would reference that history. If the LLM’s advice is only partially tailored, experts might give it a 3 on a scale of 1-5.
> - PAS ≈ Align(R_llm, U_history) = 3/5  = 0.60.
>
> 4. **Actionability & Safety Assessment (ASA)** : *Probability that a recommended action is followed*
>
> - In a pilot test with 50 sessions where the LLM recommends calling a hotline, 40 users actually do so. Then:
> - ASA ≈ P(ActionTaken | R_llm) = 40/50 = 0.80.
>
> These results indicate the LLM is empathetic (CES), culturally aware (CSI), somewhat generic in personalization (PAS), and moderately effective at promoting action (ASA). Tracking such metrics enables iterative improvements to align the model with therapeutic goals.
>
> ---
> ### **Q2.** To address your comment, we have added the following limitations to the conclusion. These are provided in two formats: a concise summary in the main manuscript (below) and a detailed version in the supplementary materials.
>
> While the SAFE-i and HAAS-e frameworks represent a significant step toward responsible and evaluative use of LLMs in mental health care, we acknowledge their limitations, which stem from the evolving nature of the field. These include 1-the rapidly evolving nature of LLM capabilities, 2-the challenge of fully capturing human emotional nuance, 3-the shifting ethical and regulatory landscape, and 4-the persistent risk of user over-reliance on AI systems. Although our frameworks address these issues through **human-in-the-loop safeguards**, empathy metrics, and escalation protocols, we recognize that continuous revision, user education, and complementary clinical oversight are essential for safe, ethical, and human-centered deployment.

---

### Official Review · Reviewer_Kohm · 2025-03-19

**Significance:** 2
**Argument Clarity:** 3
**Rating:** 3
**Confidence:** 4

**Questions:**

While many technical proposals aim to improve LLMs for healthcare, a fundamental challenge is user acceptability—how likely individuals with mental health issues are to trust and accept AI-generated responses in various personal crises. How does the paper propose strategies to effectively classify mental health issues and determine the appropriate level of support, whether from AI or a human?

The HAAS-E evaluation metrics are interesting, but they rely heavily on Align scores, which might be difficult to define consistently across different situations. How does the paper propose developing robust evaluation metrics that are both quantifiable and scalable to support efficient development?

## update after rebuttal: the authors have provided additional information to address some of my concerns. I have increased my score based on those responses.

**Discussion Potential:**

2

**Paper Summary:**

This paper advocates for LLMs as active co-creators that can work alongside humans to enhance mental health support, aiming for supportive, adaptive, and ethical values. The paper argues that this goes beyond the traditional consideration of LLMs as passive assistants in this domain. The paper offers alternative views for some concerns of LLMs for mental care on emotional intelligence limitation, over-reliance risks, and safety uncertainties. The LLMs will only be used for manageable cases, and complex cases will be escalated to human for appropriate intervention. From the analysis of prior applications of LLMs for mental health, the paper identifies three key challenges in those work, concerning the needs for ethical data foundation, robust model engineering and adaptation, and multidimensional and human-centered evaluations. The paper discusses multiple issues in those regards, such as the lack of real-word data to develop effective LLMs for mental health, the biases towards some demographics in the data, the lack of transparency of proprietary LLMs to compromise trust and reliability, and the overly focus on accuracy in current evaluation frameworks for mental health. To this end, the paper proposes the SAFE-I guidelines to position LLMs as a collaborative, adaptive, supportive, and ethical co-creators with humans in this area. The proposals focuses on real-word data collection, disparity correction, privacy protection for mental health data, high-quality data annotation, open-source model prioritization, effective model tuning and prompting, AI-human integration, personalized supports, pilot deployment, and continuous feedback. Finally, the paper presents four evaluation criteria (i.e., correctness, bias and ethical auditing, empathy, and helpfulness) along with performance metrics and validation methods for developing LLMs for mental health.

**Position:**

Yes

**Position In Title:**

Yes

**Related Work:**

3

**Strengths And Weaknesses:**

Strength:

+ This paper addresses an important and potential applications of LLMs for mental health support, which are less explored in existing work.

+ The presented analyses and challenges provide a helpful overview of current LLM-based research in mental health. In addition, the proposed guidelines and evaluation approaches offer a good starting point for future developments in this field.

+ The paper is well-structured, with each section clearly focused on a specific aspect, making it easy to read and reference.


Weakness:

- Despite the importance of these topics, a major concern with this paper is that many of the challenges, guidelines, and evaluation criteria discussed are fairly common in healthcare domains (e.g., real-world data accessibility, biases, model reliability, transparency, data annotation, multidimensional evaluation). Some also apply broadly to LLM applications across various fields (e.g., prompting and tuning, human-AI integration, personalization). The novelty of the proposals is thus limited. It will be more convincing if the paper presents more specific challenges for mental health support, and propose corresponding solutions for those challenges.

- The human-centered evaluation of LLMs for this domain is helpful. However, the proposed approach of continuous user feedback, especially in critical cases, seems less practical for mental health support. Ultimately, when someone with a mental health issue seeks help, they are likely focused on their immediate concerns rather than providing real-time feedback on the system. The reliability of feedback in those cases is also unclear.  Addressing these unique challenges would strengthen the paper.

**Support:**

3

---

> ### Author Rebuttal · Authors · 2025-04-01
>
> ### **W1.**
> #### We appreciate the reviewer’s comment and agree that several challenges we discuss are also relevant to broader healthcare and LLM applications. We want to clarify that our positions are strongly informed by extensive practical experience collaborating with a leading mental health organization. During the development and evaluation phases of our LLM-based support systems in real-world mental health crisis settings, we encountered several critical, mental-health-specific challenges that differed from the general healthcare we propose in this paper [REF] (blind review). We would like to emphasize that the problems of data availability, modeling approaches, and evaluation frameworks in mental health are distinct and underrepresented in current AI literature, as highlighted in our introduction.
> ---
> ### **1. Comparison Table**
> #### We provided a clear comparison between the general health and mental health domains, which are now included as a table in the supplementary materials. Due to rebuttal word limits, only a few rows are included here and also references will be added to the main manuscript:
>
> **Table 1: Mental Health vs. General Health LLM Applications**
>
> | **Component**      | **Mental Health LLMs** | **General Health LLMs** | **Mental Health Specific Challenge**                  |
> |--------------------|-------------|-------------|--------------------------------------------|
> | Real World Data          | Essential   | Important   | Limited availability of real-world conversational data for the mental health domain compared to the health domain [1].  |
> | Empathy and Therapeutic Assessment      | Essential   | Minimal     | Explicit evaluation of therapeutic alliance and genuine emotional empathy is critical [2].|
> | Personalization Appropriateness Score | Essential   | Moderate    | Personalization tailored to emotional and psychological profiles is necessary [3].        |
>
> #### **Ranking Importance:** When ranking the importance of each criterion (Essential, Important, Moderate, Minimal), we used a qualitative scale similar to those in prior research [4].
> ---
> #### **2. New Text in Introduction** We added a clear paragraph to the Introduction to highlight the mental health organization collaboration, the difference between LLM in health and mental health, and a reference to the table.
> ---
>
> #### **W2.** Our goal is not to burden users during critical moments but to propose a structured, multi-method evaluation approach that includes:
> - **Post-interaction feedback:** Collected retrospectively and voluntarily, minimizing burden and reliability concerns.
> - **Expert-driven evaluation (e.g., red-teaming, RCTs):** Conducted by trained clinicians to identify risks proactively.
> - **Stakeholder-informed iterative refinement:** Involves crisis responders and clinicians in post-service feedback sessions.
>
> #### As noted in **Sections 5 and 6** of our paper, we already address this concern by advocating structured human evaluations and post-interaction mechanisms.
> ---
>
> #### **Q1.** We appreciate the reviewer’s concern. Our paper addresses user acceptability through both **SAFE-i** and **HAAS-e** frameworks:
>
> #### **1. Building Trust and Acceptability**
> - **Transparency & Explainability:** Users are informed when and why AI is used (Section 5.3).
> - **Human-AI Complementarity:** AI acts as a co-creator, not a replacement; high-risk cases are escalated to human responders (Sections 5.3 & 6.1).
> - **Risk-Based Escalation:** Clear triggers for human intervention (e.g., self-harm) (Section 5.2).
>
> #### **2. New Literature on AI Acceptance in Mental Health** User acceptability underpins the effectiveness of AI-driven mental health interventions. Recent studies indicate a positive public perception towards AI in mental health contexts. Varghese et al. reported that approximately 53% of respondents expressed moderate trust in AI-driven mental health interventions, appreciating AI's accessibility, anonymity, and reduced stigma [1]. Similarly, Alanezi highlighted that patients positively perceived ChatGPT for psychoeducation, emotional support, and practical guidance, indicating a growing trust and acceptance [2].
> ---
> #### **Q2.**
> #### **1. New Use Case in Supplementary Materials**
> We added a detailed example demonstrating how to operationalize robust and scalable evaluation metrics. This is also referenced in our response to Reviewer 2 (H2Hm).
>
> #### **2. Practical Strategies for Evaluation**
> - **Beyond AlignScore:** Combine CES, CSI, PAS, and ASA with qualitative interviews, stakeholder feedback, and clinical review.
> - **LLM-as-a-Judge:** Expert-instructed LLMs score alignment by comparing model output to human annotations.
> - **Log-likelihood:** Compare preferred vs. rejected responses using model probabilities.
> - **Expert-in-the-loop:** Validate alignment metrics with expert judgment in high-risk scenarios.

---

> > ### Comment · Reviewer_Kohm · 2025-04-02
> >
> > Thank you for your responses. While I still have some concerns as detailed in my review, I understand the challenges to work on this area, and appreciate the collaborative efforts with mental health organizations. I updated my score in light of the rebuttal.

---

> > > ### Author Response · Authors · 2025-04-03
> > >
> > > Thank you for updating your evaluation and for considering our responses. We also appreciate your acknowledgment of the challenges inherent in this research area, and we will make sure to address your comments further in our future work.

---

### Official Review · Reviewer_qmmD · 2025-03-28

**Significance:** 2
**Argument Clarity:** 2
**Rating:** 2
**Confidence:** 4

**Questions:**

1. Could you provide further details or preliminary findings on the HAAS-E Evaluation Framework, specifically regarding the quantification of metrics such as the Cultural Sensitivity Index and the Contextual Empathy & Emotional Score?
2. How will you address the ethical and privacy challenges related to collecting and using sensitive mental health data while adhering to regulatory standards such as HIPAA and GDPR?
3. It is confusing that the title says LLMs as co-creators, but it claims to implement LLMs as complementary tools rather than replacements for human therapists. So what are the co-creators for?

**Discussion Potential:**

2

**Paper Summary:**

The paper argues for a paradigm shift in integrating LLMs into mental health care. Rather than being seen as mere assistive tools, LLMs should be reimagined as ethical, adaptive co-creators that actively enhance human-led interventions. To achieve this, the authors introduce two complementary frameworks: the SAFE-I Implementation Guidelines, which provide a blueprint for ethical data practices, adaptive model engineering, and real-world integration, and the HAAS-E Evaluation Framework, which proposes multidimensional metrics, such as encompassing trustworthiness, empathy, cultural sensitivity, and actionability, to assess LLM performance in mental health contexts.

**Position:**

Yes

**Position In Title:**

Yes

**Related Work:**

3

**Strengths And Weaknesses:**

Strengths:
1. The paper conducts a comprehensive review of the challenges and counterpositions related to deploying LLMs in mental health, introduces an innovative dual-framework approach (SAFE-I and HAAS-E), and focuses on ethical, culturally sensitive, and personalized care.
2. The structured guidelines provide a clear roadmap for mitigating risks like bias, dehumanization, and regulatory uncertainty, which are highly relevant topics for the ICML community.
3. The emphasis on multidimensional evaluation beyond traditional technical metrics is likely to inspire robust discussion among researchers and practitioners.

Weaknesses:
1. The position seems overly ambitious due to the complexity of operationalizing these frameworks in diverse clinical settings.
2. There is also a need for more empirical evidence or pilot studies demonstrating the feasibility of the approach, particularly regarding the quantification of nuanced metrics like cultural sensitivity and empathy.

Suggestions for improvement include:
1. Detailing implementation strategies for resource-constrained environments
2. Clarifying how these frameworks will be integrated with existing mental health infrastructures.

**Support:**

3

---

> ### Author Rebuttal · Authors · 2025-04-01
>
> ### **W1. Response**
>
> We appreciate the reviewer’s insightful comment. Below is a three-point summary of the proposed framework's main strategy to address the complexity of operationalizing:
>
> 1. **Grounded Practical Collaboration**:
>    We want to clarify that our position is grounded in extensive, hands-on collaboration with a leading mental health organization. While developing and evaluating our LLM-based support systems in real-world crisis settings, we faced—and addressed—unique, domain-specific challenges that extend beyond typical healthcare contexts [REF].
>
> 2. **Addressing the Ambition of SAFE-i and HAAS-e**:
>    We acknowledge the concern regarding the ambitious scope of operationalizing SAFE-i and HAAS-e across diverse clinical environments. However, this ambition is precisely what drives our contribution: we offer structured pathways covering ethical data governance, adaptive engineering, and human-centered evaluation that directly address the complexities of real-world LLM deployment. We provide an optimal scenario as a guideline, recognizing that not all components will be fully implemented in every setting.
>
> 3. **Evidence-Based Guidelines and Refinement**:
>    Lastly, our prior work demonstrates that structured guidelines and iterative refinement are needed in actual mental health settings. We therefore believe our frameworks tackle the gap we found in previously proposed systems—ensuring LLMs can be responsibly scaled and adapted to varying clinical contexts.
>
> > **[REF]** The paper on our collaboration with a mental health institute. We will add it if the paper is accepted to preserve the blind review process.
>
> ---
>
> ### **W2 and Q1. Response**
>
> Thank you for your feedback, and we agree that a clear example should be included in the HAAS-E Evaluation Framework. To strengthen the clarity and practicality of our proposal, we have included an example use case in our response to Reviewer 2 (H2Hm). It demonstrates how we develop robust evaluation metrics that are both **quantifiable** and **scalable** to support efficient development. We will also include this example in the camera-ready version of our paper.
>
> ---
>
> ### **Q2. Privacy and Regulatory Compliance**
>
> We appreciate the reviewer’s question about safeguarding sensitive mental health data. Below, we provide specific privacy details to be added to the **Compliance with Regulatory Standards** subsection:
>
> - **Privacy-by-Design**:
>   Our framework mandates that sensitive data be de-identified and encrypted from the outset, ensuring no personally identifiable information is visible during training or evaluation. This approach not only meets but often exceeds HIPAA and GDPR requirements.
>
> - **Strict Access Control**:
>   We implement role-based permissions and secure storage protocols—encrypting data both in transit and at rest—and routinely conduct audits to ensure only authorized personnel or systems can handle sensitive information. All sensitive information is removed before LLM training.
>
> - **Transparent Consent**:
>   We incorporate explicit opt-in and opt-out mechanisms. Users are informed about how and why their data is used and retain the ability to withdraw consent without affecting their access to care.
>
> ---
>
> ### **Q3. On "Co-Creation" Terminology**
>
> We thank the reviewer for highlighting this. The term **"co-creators"** emphasizes **collaborative AI-human interactions** rather than autonomy or replacement of human therapists.
>
> Co-creation is essential to achieving this complementarity because it ensures AI tools are developed with the people who will use and be affected by them—clinicians, patients, ethicists, and health system leaders. Through co-creation, we integrate human values, domain expertise, and lived experiences into the design and training of AI systems.
>
> This process not only helps tailor models to real-world needs, but also fosters trust, transparency, and usability. It leads to AI that augments (rather than undermines) clinical workflows, supports decision-making under uncertainty, and adapts to human cognitive and emotional contexts. In doing so, co-creation becomes a foundational method for aligning AI systems with human strengths, closing the gap between algorithmic potential and meaningful health impact.

---

### Decision · Program_Chairs · 2025-04-30

**Decision:**

Accept (poster)

**Comment:**

This paper makes a strong case for the need for AI to fill a gap in the mental healthcare space as an augmentative tool. Reviewers identified the comprehensive nature of the paper to be a strength as the paper presents a thorough discussion of the issues of having a LLM fill such a role in mental healthcare. In addition, reviewers felt the paper provided practical guidelines for achieving the proposed position (e.g. the evaluation metrics and empirical validation methods), though some parts needed further clarification.

The main criticism of this work is that it is a very ambitious vision. Reviewers pointed out that some aspects of the vision (e.g. developing cultural sensitivity and empathy, developing trust, obtaining sensitive mental health data) are very challenging and they questioned the feasibility of achieving these requirements.

The paper definitely fits the criterion of a position paper that would stimulate discussion on timely topics that need our community's input.